# Tight Dimension Independent Lower Bound on the Expected Convergence Rate for Diminishing Step Sizes in SGD

**Phuong Ha Nguyen**
Electrical and Computer Engineering
University of Connecticut, USA
phuongha.ntu@gmail.com

**Lam M. Nguyen**
IBM Research, Thomas J. Watson Research Center
Yorktown Heights, USA
LamNguyen.MLTD@ibm.com

**Marten van Dijk**
Electrical and Computer Engineering
University of Connecticut, USA
marten.van_dijk@uconn.edu

## Abstract

We study the convergence of Stochastic Gradient Descent (SGD) for strongly convex objective functions. We prove for all $t$ a lower bound on the expected convergence rate after the $t$-th SGD iteration; the lower bound is over all possible sequences of diminishing step sizes. It implies that recently proposed sequences of step sizes at ICML 2018 and ICML 2019 are *universally* close to optimal in that the expected convergence rate after *each* iteration is within a factor 32 of our lower bound. This factor is independent of dimension $d$. We offer a framework for comparing with lower bounds in state-of-the-art literature and when applied to SGD for strongly convex objective functions our lower bound is a significant factor $775 \cdot d$ larger compared to existing work.

## 1 Introduction

We are interested in solving the following stochastic optimization problem

$$\min_{w \in \mathbb{R}^d} \{F(w) = \mathbb{E}[f(w; \xi)]\}, \tag{1}$$

where $\xi$ is a random variable obeying some distribution $g(\xi)$. In the case of empirical risk minimization with a training set $\{(x_i, y_i)\}_{i=1}^n$, $\xi_i$ is a random variable that is defined by a single random sample $(x, y)$ pulled uniformly from the training set. Then, by defining $f_i(w) := f(w; \xi_i)$, empirical risk minimization reduces to

$$\min_{w \in \mathbb{R}^d} \left\{F(w) = \frac{1}{n} \sum_{i=1}^n f_i(w)\right\}. \tag{2}$$

Problems of this type arise frequently in supervised learning applications [8]. The classic first-order methods to solve problem (2) are gradient descent (GD) [19] and stochastic gradient descent (SGD)[1] [21] algorithms. GD is a standard deterministic gradient method, which updates iterates along the negative full gradient with learning rate $\eta_t$ as follows

$$w_{t+1} = w_t - \eta_t \nabla F(w_t) = w_t - \frac{\eta_t}{n} \sum_{i=1}^n \nabla f_i(w_t) , \ t \geq 0.$$

We can choose $\eta_t = \eta = \mathcal{O}(1/L)$ and achieve a linear convergence rate for the strongly convex case [15]. The upper bound of the convergence rate of GD and SGD has been studied in [2, 4, 15, 22, 17, 16, 7].

The disadvantage of GD is that it requires evaluation of $n$ derivatives at each step, which is very expensive and therefore avoided in large-scale optimization. To reduce the computational cost for solving (2), a class of variance reduction methods [11, 5, 9, 18] has been proposed. The difference between GD and variance reduction methods is that GD needs to compute the full gradient at each step, while the variance reduction methods will compute the full gradient after a certain number of steps. In this way, variance reduction methods have less computational cost compared to GD. To avoid evaluating the full gradient at all, SGD generates an unbiased random variable $\xi_t$ satisfying

$$\mathbb{E}_{\xi_t}[\nabla f(w_t; \xi_t)] = \nabla F(w_t),$$

and then evaluates gradient $\nabla f(w_t; \xi_t)$ for $\xi_t$ drawn from a distribution $g(\xi)$. After this, $w_t$ is updated as follows

$$w_{t+1} = w_t - \eta_t \nabla f(w_t; \xi_t). \tag{3}$$

We focus on the general problem (1) where $F$ is strongly convex. Since $F$ is strongly convex, a unique optimal solution of (1) exists and throughout the paper we denote this optimal solution by $w_*$ and are interested in studying the expected convergence rate

$$Y_t = \mathbb{E}[\|w_t - w_*\|^2].$$

Algorithm 1 provides a detailed description of SGD. Obviously, the computational cost of a single iteration in SGD is $n$ times cheaper than that of a single iteration in GD. However, as has been shown in literature we need to choose $\eta_t = \mathcal{O}(1/t)$ and the expected convergence rate of SGD is slowed down to $\mathcal{O}(1/t)$ [3], which is a sublinear convergence rate.

---

**Algorithm 1** Stochastic Gradient Descent (SGD) Method

---

  **Initialize:** $w_0$
  **Iterate:**
  **for** $t = 0, 1, \ldots$ **do**
    Choose a step size (i.e., learning rate) $\eta_t > 0$.
    Generate a random variable $\xi_t$ with probability density $g(\xi_t)$.
    Compute a stochastic gradient $\nabla f(w_t; \xi_t)$.
    Update the new iterate $w_{t+1} = w_t - \eta_t \nabla f(w_t; \xi_t)$.
  **end for**

---

**Problem Statement and Contributions:** We seek to find a tight lower bound on the expected convergence rate $Y_t$ with the purpose of showing that the stepsize sequences of [17] and [7] for classical SGD is optimal for $\mu$-strongly convex and $L$-smooth respectively expected $L$-smooth objective functions within a *small dimension independent constant factor*. This is important because of the following reasons:

1. The lower bound tells us that a sequence of stepsizes as a function of only $\mu$ and $L$ cannot beat an expected convergence rate of $O(1/t)$ – this is known general knowledge and was already proven in [1], where a *dimension dependent* lower bound for a larger class of algorithms that includes SGD was proven. For the class of SGD with diminishing stepsizes as a function of only global parameters $\mu$ and $L$ we show a *dimension independent* lower bound which is a factor $775 \cdot d$ larger.

2. We now understand into what extent the sequence of stepsizes of [17] and [7] are optimal in that it leads to minimal expected convergence rates $Y_t$ for *all* $t$: For each $t$ we will show a *dimension independent* lower bound on $Y_t$ over *all possible* stepsize sequences. This includes the *best possible* stepsize sequence which minimizes $Y_t$ for a *given* $t$. Our lower bound achieves the upper bound on $Y_t$ for the stepsize sequences of [17] and [7] within a factor 32 for *all* $t$. This implies that these stepsize sequences universally minimizes each $Y_t$ within factor 32.

3. As a consequence, in order to attain a better expected convergence rate, we need to *either* assume more specific knowledge about the objective function $F$ so that we can construct a better stepsize sequence for SGD based on this additional knowledge *or* we need to step away from SGD and use a different kind of algorithm. For example, the larger class of algorithms in [1] may contain a non-SGD algorithm which may get close to the lower bound proved in [1] which is a factor $775 \cdot d$ smaller. Since the larger class of algorithms in [1] contains algorithms such as Adam [10], AdaGrad [6], SGD-Momentum [23], RMSProp [24] we now know that these practical algorithms will at most improve a factor $32 \cdot 775 \cdot d$ over SGD for strongly convex optimization – this can be significant as this can lead to orders of magnitude less gradient computations. We are the first to make such quantification.

**Outline:** Section 2 discusses background: First, we discuss the recurrence on $Y_t$ used in [17] for proving their upper bound on $Y_t$ – this recurrence plays a central role in proving our lower bound. We discuss the upper bounds of both [17] and [7] – the latter holding for a larger class of algorithms. Second, we explain the lower bound of [1] in detail in order to be able to properly compare with our lower bound. Section 3 introduces a framework for comparing bounds and explains the consequences of our lower bound in detail. Section 4 describes a class of strongly convex and smooth objective functions which is used to derive our lower bound. We also verify our theory by experiments in the supplementary material. Section 5 concludes the paper.

## 2 Background

We explain the upper bound of [17, 7], and lower bound of [1] respectively.

### 2.1 Upper Bound for Strongly Convex and Smooth Objective Functions

The starting point for analysis is the recurrence first introduced in [17, 12]

$$\mathbb{E}[\|w_{t+1} - w_*\|^2] \leq (1 - \mu\eta_t)\mathbb{E}[\|w_t - w_*\|^2] + \eta_t^2 N, \tag{4}$$

where

$$N = 2\mathbb{E}[\|\nabla f(w_*; \xi)\|^2]$$

and $\eta_t$ is upper bounded by $\frac{1}{2L}$; the recurrence has been shown to hold, see [17, 12], if we assume

1. $F(.)$ is $\mu$-strongly convex,

2. $f(w; \xi)$ is $L$-smooth,

3. $f(w; \xi)$ is convex, and

4. $N$ is finite;

we detail these assumptions below:

**Assumption 1** ($\mu$-strongly convex). *The objective function $F : \mathbb{R}^d \to \mathbb{R}$ is $\mu$-strongly convex, i.e., there exists a constant $\mu > 0$ such that $\forall w, w' \in \mathbb{R}^d$,*

$$F(w) - F(w') \geq \langle \nabla F(w'), (w - w') \rangle + \frac{\mu}{2}\|w - w'\|^2. \tag{5}$$

**Assumption 2** ($L$-smooth). *$f(w; \xi)$ is $L$-smooth for every realization of $\xi$, i.e., there exists a constant $L > 0$ such that, $\forall w, w' \in \mathbb{R}^d$,*

$$\|\nabla f(w; \xi) - \nabla f(w'; \xi)\| \leq L\|w - w'\|. \tag{6}$$

Assumption 2 implies that $F$ is also $L$-smooth.

**Assumption 3.** *$f(w; \xi)$ is convex for every realization of $\xi$, i.e., $\forall w, w' \in \mathbb{R}^d$,*

$$f(w; \xi) - f(w'; \xi) \geq \langle \nabla f(w'; \xi), (w - w') \rangle.$$

**Assumption 4.** *$N = 2\mathbb{E}[\|\nabla f(w_*; \xi)\|^2]$ is finite.*

We denote the set of strongly convex objective functions by $\mathcal{F}_{str}$ and denote the subset of $\mathcal{F}_{str}$ satisfying Assumptions 1, 2, 3, and 4 by $\mathcal{F}_{sm}$.

We notice that the earlier established recurrence in [13] under the same set of assumptions

$$\mathbb{E}[\|w_{t+1} - w_*\|^2] \leq (1 - 2\mu\eta_t + 2L^2\eta_t^2)\mathbb{E}[\|w_t - w_*\|^2] + \eta_t^2 N$$

is similar, but worse than (4) as it only holds for $\eta_t < \frac{\mu}{L^2}$ where (4) holds for $\eta_t \leq \frac{1}{2L}$. Only for step sizes $\eta_t < \frac{\mu}{2L^2}$ the above recurrence provides a better bound than (4), i.e., $1 - 2\mu\eta_t + 2L^2\eta_t^2 \leq 1 - \mu\eta_t$. In practical settings such as logistic regression $\mu = \mathcal{O}(1/n)$, $L = \mathcal{O}(1)$, and $t = \mathcal{O}(n)$ (i.e. $t$ is at most a relatively small constant number of epochs, where a single epoch represents $n$ iterations resembling the complexity of a single GD computation). See (8) below, for this parameter setting the optimally chosen step sizes are $\gg \frac{\mu}{L^2}$. This is the reason we focus in this paper on analyzing recurrence (4) in order to prove our lower bound: For $\eta_t \leq \frac{1}{2L}$,

$$Y_{t+1} \leq (1 - \mu\eta_t)Y_t + \eta_t^2 N, \tag{7}$$

where $Y_t = \mathbb{E}[\|w_t - w_*\|^2]$.

Based on the above assumptions (*without* the so-called bounded gradient assumption) and knowledge of only $\mu$ and $L$ a sequence of step sizes $\eta_t$ can be constructed such that $Y_t$ is smaller than $\mathcal{O}(1/t)$ [17]; more explicitly, for the sequence of step sizes

$$\eta_t = \frac{2}{\mu t + 4L} \tag{8}$$

we have for all objective functions in $\mathcal{F}_{sm}$ the upper bound

$$Y_t \leq \frac{16N}{\mu} \frac{1}{\mu(t - T') + 4L} = \frac{16N}{\mu^2 t}(1 + \mathcal{O}(1/t)), \tag{9}$$

where

$$t \geq T' = \frac{4L}{\mu} \max\{\frac{L\mu Y_0}{N}, 1\} - \frac{4L}{\mu}.$$

We notice that [7] studies the larger class, which we denote $\mathcal{F}_{esm}$, which is defined as $\mathcal{F}_{sm}$ where expected smoothness is assumed in stead of smoothness and convexity of component functions. We rephrase their assumption for classical SGD as studied in this paper.[2]

**Assumption 5.** *(L-smooth in expectation) The objective function* $F : \mathbb{R}^d \to \mathbb{R}$ *is L-smooth in expectation if there exists a constant* $L > 0$ *such that,* $\forall w \in \mathbb{R}^d$,

$$\mathbb{E}[\|\nabla f(w; \xi) - \nabla f(w_*; \xi)\|^2] \leq 2L\|F(w) - F(w_*)\|. \tag{10}$$

The results in [7] assume the above assumption for empirical risk minimization (2). $L$-smoothness, see [15], implies Lipschitz continuity (i.e., $\forall w, w' \in \mathbb{R}^d$,

$$f(w, \xi) \leq f(w', \xi) + \langle \nabla f(w', \xi), (w - w') \rangle + \frac{L}{2}\|w - w'\|^2$$

) and together with Proposition A.1 in [7] this implies $L$-smooth in expectation. This shows that $\mathcal{F}_{esm}$ defined by Assumptions 1, 4, and 5 is indeed a superset of $\mathcal{F}_{sm}$.

The step sizes (8) from [17] for $\mathcal{F}_{sm} \subseteq \mathcal{F}_{esm}$ and

$$\eta_t = \frac{2t + 1}{(t + 1)^2 \mu} \text{ for } t > \frac{4L}{\mu} \text{ and } \eta_t = \frac{1}{2L} \text{ for } t \leq \frac{4L}{\mu} \tag{11}$$

developed for $\mathcal{F}_{esm}$ in [7] and [17] are equivalent in that they are both $\approx \frac{2}{\mu t}$ for $t$ large enough. Both step size sequences give exactly the same asymptotic upper bound (9) on $Y_t$ (in our notation).

In [21], the authors proved the convergence of SGD for the step size sequence $\{\eta_t\}$ satisfying conditions $\sum_{t=0}^{\infty} \eta_t = \infty$ and $\sum_{t=0}^{\infty} \eta_t^2 < \infty$. In [13], the authors studied the expected convergence rates for another class of step sizes of $\mathcal{O}(1/t^p)$ where $0 < p \leq 1$. However, the authors of both [21] and [13] do *not* discuss about the optimal step sizes among all proposed step sizes which is what is done in this paper.

## 2.2 Lower Bound for First Order Stochastic Oracles

The authors of [14] proposed the first formal study on lower bounding the expected convergence rate for a large class of algorithms which includes SGD. The authors of [1] and [20] independently studied this lower bound using information theory and were able to improve it.

The derivation in [1] is for algorithms including SGD where the sequence of stepsizes is a-priori fixed based on global information regarding assumed stochastic parameters concerning the objective function $F$. Their proof uses the following set of assumptions: First, The assumption of a strongly convex objective function, i.e., Assumption 1 (see Definition 3 in [1]). Second, the objective function is convex Lipschitz:

**Assumption 6.** *(convex Lipschitz) The objective function $F$ is a convex Lipschitz function, i.e., there exists a bounded convex set $\mathcal{S} \subset \mathbb{R}^d$ and a positive number $K$ such that $\forall w, w' \in \mathcal{S} \subset \mathbb{R}^d$*

$$\|F(w) - F(w')\| \le K\|w - w'\|.$$

We notice that this assumption implies the assumption on bounded gradients as stated here (and explicitly mentioned in Definition 1 in [1]): There exists a bounded convex set $\mathcal{S} \subset \mathbb{R}^d$ and a positive number $\sigma$ such that

$$\mathbb{E}[\|\nabla f(w;\xi)\|^2] \le \sigma^2 \tag{12}$$

for all $w \in \mathcal{S} \subset \mathbb{R}^d$. This is not the same as the bounded gradient assumption where $\mathcal{S} = \mathbb{R}^d$ is unbounded.[3] Clearly, for $w_*$, (12) implies a finite $N \le 2\sigma^2$.

We define $\mathcal{F}_{lip}$ as the set of strongly convex objective functions that satisfy Assumption 6. Classes $\mathcal{F}_{esm}$ and $\mathcal{F}_{lip}$ are both subsets of $\mathcal{F}_{str}$ and differ (are not subclasses of each other) in that they assume expected smoothness and convex Lipschitz respectively.

To prove a lower bound of $Y_t$ for $\mathcal{F}_{lip}$, the authors constructed a class of objective functions $\subseteq \mathcal{F}_{lip}$ and showed a lower bound of $Y_t$ for this class; in terms of the notation used in this paper,

$$\frac{\log(2/\sqrt{e})}{432 \cdot d} \frac{N}{\mu^2 t}. \tag{13}$$

The authors of [1] prove lower bound (13) for the class $\mathcal{A}_{stoch}$ of *stochastic first order algorithms* that can be understood as operating based on information provided by a stochastic first-order oracle, i.e., any algorithm which bases its computation in the $t$-th iteration on $\mu$, $K$ or $L$, $d$, and access to an oracle that provides $f(w_t;\xi_t)$ and $\nabla f(w_t;\xi_t)$. This class includes $\mathcal{A}_{SGD}$ defined as SGD with some sequence of diminishing step sizes as a function of global parameters such as $\mu$ and $L$ or $\mu$ and $K$, see Algorithm 1. We notice that $\mathcal{A}_{stoch}$ also includes practical algorithms such as Adam [10], etc. We revisit their derivation in the supplementary material where we show[4] how their lower bound transforms into (13). Notice that their lower bound depends on dimension $d$.

## 3 Framework for Upper and Lower Bounds

Let $par(F)$ denote the concrete values of the global parameters of an objective function $F$ such as the values for $\mu$ and $L$ corresponding to objective functions $F$ in $\mathcal{F}_{sm}$ and $\mathcal{F}_{esm}$ or $\mu$ and $K$ corresponding to objective functions $F$ in $\mathcal{F}_{lip}$. When defining a class $\mathcal{F}$ of objective functions, we also need to explain how $\mathcal{F}$ defines a corresponding $par(.)$ function. We will use the notation $\mathcal{F}[p]$ to stand for the subclass $\{F \in \mathcal{F} : p = par(F)\} \subseteq \mathcal{F}$, i.e., the subclass of objective functions of $\mathcal{F}$ with the same parameters $p$. We assume that parameters of a class are included in the parameters of a smaller subclass: For example, $\mathcal{F}_{sm}$ is a subset of the class of strongly convex objective functions $\mathcal{F}_{str}$ with only global parameter $\mu$. This means that for concrete values $\mu$ and $L$ we have $\mathcal{F}_{sm}[\mu, L] \subseteq \mathcal{F}_{str}[\mu]$.

For a given objective function $F$, we are interested in the best possible expected convergence rate after the $t$-th iteration among all possible algorithms $A$ in a larger class of algorithms $\mathcal{A}$. Here, we

assume that $\mathcal{A}$ is a subclass of the larger class $\mathcal{A}_{stoch,\mathcal{U}}$ of stochastic first order algorithms where the computation in the $t$-th iteration not only has access to $par(F)$ and access to an oracle that provides $f(w_t; \xi_t)$ and $\nabla f(w_t; \xi_t)$ but also access to possibly another oracle $\mathcal{U}$ providing even more information. Notice that $\mathcal{A} \subseteq \mathcal{A}_{stoch} \subseteq \mathcal{A}_{stoch,\mathcal{U}}$ for any oracle $\mathcal{U}$. With respect to the expected convergence rate, we want to know which algorithm $A$ in $\mathcal{A}$ minimizes $Y_t$ the most. Notice that for different $t$ this may be a different algorithm $A$. We define for $F \in \mathcal{F}$ (with associated $par(.)$)

$$\gamma_t^F(\mathcal{A}) = \inf_{A \in \mathcal{A}} Y_t(F, A),$$

where $Y_t$ is explicitly shown as a function of the objective function $F$ and choice of algorithm $A$.

Among the objective functions $F \in \mathcal{F}$ with same global parameters $p = par(F)$ (i.e., $F \in \mathcal{F}[p]$), we consider the objective function $F$ which has the *worst* expected convergence rate at the $t$-th iteration. This is of interest to us because algorithms $A$ only have access to $p = par(F)$ as the sole information about objective function $F$, hence, if we prove an upper bound on the expected convergence rate for algorithm $A$, then this upper bound must hold for all $F \in \mathcal{F}$ with the same parameters $p = par(F)$. In other words such an upper bound must be at least

$$\gamma_t(\mathcal{F}[p], \mathcal{A}) = \sup_{F \in \mathcal{F}[p]} \gamma_t^F(\mathcal{A}) = \sup_{F \in \mathcal{F}[p]} \inf_{A \in \mathcal{A}} Y_t(F, A).$$

So, any lower bound on $\gamma_t(\mathcal{F}[p], \mathcal{A})$ gives us a lower bound on the best possible upper bound on $Y_t$ that can be achieved. Such a lower bound tells us into what extent the expected convergence rate $Y_t$ cannot be improved.

The lower bound (13) and upper bound (9) are not only a function of $\mu$ in $p = par(F)$ but also a function of $N$ which is outside $p = par(F)$ for $F \in \mathcal{F}_{lip}$ or $F \in \mathcal{F}_{esm}$. We are really interested in such more fine-grained bounds that are a function of $N$. For this reason we need to consider the subclass of objective functions $F$ in $\mathcal{F}[p]$ that all have the same $N$. We implicitly understand that $N$ is an auxiliary parameter of an objective function $F$ and we denote this as a function of $F$ as $N(F)$. We define $\mathcal{F}^a[p] = \{F \in \mathcal{F}[p] \ : \ a = aux(F)\}$ where $aux(.)$ represents for example $N(.)$. This leads to notation like $\mathcal{F}_{lip}^N[\mu, K, d]$. Notice that $p = par(F)$ can be used by an algorithm $A \in \mathcal{A}$ while $a = aux(F)$ is not available to $A$ through $p = par(F)$ (but may be available through access to an oracle).

If we find a tight lower bound with upper bound up to a constant factor, as in this paper, then we know that the algorithm that achieves the upper bound is close to optimal in that the expected convergence rate cannot be further minimized/improved in a significant way. In practice we are only interested in upper bounds on $Y_t$ that can be realized by the *same* algorithm $A$ (if not, then we need to know a-priori the exact number of iterations $t$ we want to run an algorithm and then choose the best one for that $t$). In this paper we consider the algorithm $A$ for $F$ in $\mathcal{F}_{sm}$ resp. $\mathcal{F}_{esm}$ defined as SGD with diminishing step sizes (8) resp. (11) as a function of $par(F) = (\mu, L)$ giving upper bound (9) on expected convergence rate $Y_t(F, A)$. We show that $A$ is close to optimal.

Given the above definitions we have

$$\gamma_t(\mathcal{F}[p], \mathcal{A}) \leq \gamma_t(\mathcal{F}'[p'], \mathcal{A}') \tag{14}$$

for $\mathcal{F}[p] \subseteq \mathcal{F}'[p']$ and $\mathcal{A}' \subseteq \mathcal{A}$, i.e., the worst objective function in a larger class of objective functions is worse than the worst objective function in a smaller class of objective functions (see the supremum used in defining $\gamma_t$) and the best algorithm from a larger class of algorithms is better than the best algorithm from a smaller class of algorithms (see the infimum used in defining $\gamma_t$). This implies

$$\gamma_t(\mathcal{F}_{lip}^N[\mu, K, d], \mathcal{A}_{stoch}) \quad \leq \quad \gamma_t(\mathcal{F}_{str}^N[\mu], \mathcal{A}_{SGD}), \tag{15}$$

$$\gamma_t(\mathcal{F}_{sm}^N[\mu, L], \mathcal{A}_{ExtSGD}) \quad \leq \quad \gamma_t(\mathcal{F}_{esm}^N[\mu, L], \mathcal{A}_{SGD}) \leq \gamma_t(\mathcal{F}_{str}^N[\mu], \mathcal{A}_{SGD}), \tag{16}$$

where $\mathcal{A}_{SGD} \subseteq \mathcal{A}_{ExtSGD}$ is defined as follows:

In our framework we introduce *extended SGD* as the class $\mathcal{A}_{ExtSGD}$ of SGD algorithms where the stepsize in the $t$-th iteration can be computed based on global parameters $\mu$, $L$, and access to an oracle $\mathcal{U}$ that provides additional information $N$, $\nabla F(w_t)$, and $Y_t$. This class also includes SGD with

diminishing stepsizes as defined in Algorithm 1, i.e., $\mathcal{A}_{SGD} \subseteq \mathcal{A}_{ExtSGD}$. The reason for introducing the larger class $\mathcal{A}_{ExtSGD}$ is not because it contains practical algorithms different than SGD, on the contrary. The only reason is that it allows us to define *one single algorithm* $A \in \mathcal{A}_{ExtSGD}$ which realizes $\gamma_t^F(\mathcal{A}_{ExtSGD})$ *for all $t$* for all $F$ in a to be constructed subclass $\mathcal{F} \subseteq \mathcal{F}_{sm}$ – the topic of the next section. This property allows a rather straightforward calculus based proof without needing to use more advanced concepts from information and probability theory as required in the proof of [1]. Looking ahead, we will prove in Theorem 1

$$\frac{1}{2}\frac{N}{\mu^2 t}(1 - \mathcal{O}((\ln t)/t)) \leq \gamma_t(\mathcal{F}_{sm}^N[\mu, L], \mathcal{A}_{ExtSGD}). \tag{17}$$

Notice that the construction of $\eta_t$ for algorithms in $\mathcal{A}_{ExtSGD}$ does *not* depend on knowledge of the stochastic gradient $\nabla f(w_t; \xi_t)$. So, we do not consider step sizes that are adaptively computed based on $\nabla f(w_t; \xi_t)$.

As a disclaimer we notice that for some objective functions $F \in \mathcal{F}_{sm}^N[\mu, L]$ the expected convergence rate can be much better than what is stated in (17); this is because $\gamma_t(\{F\}, \mathcal{A}_{ExtSGD})$ can be much smaller than $\gamma_t(\mathcal{F}_{sm}^N[\mu, L], \mathcal{A}_{ExtSGD})$, see (14). This is due to the specific nature of the objective function $F$ itself. However, without knowledge about this nature, one can only prove a general upper bound on the expected convergence rate $Y_t$ and any such upper bound must be at least the lower bound (17).

Results (13) and (9) of the previous section combined with (15), (16), and (17) yield

$$\frac{\log(2/\sqrt{e})}{432 \cdot d}\frac{N}{\mu^2 t} \leq \gamma_t(\mathcal{F}_{lip}^N[\mu, K, d], \mathcal{A}_{stoch}) \quad \leq \quad \gamma_t(\mathcal{F}_{str}^N[\mu], \mathcal{A}_{SGD}), \tag{18}$$

$$\frac{1}{2}\frac{N}{\mu^2 t}(1 - \mathcal{O}((\ln t)/t)) \leq \gamma_t(\mathcal{F}_{esm}^N[\mu, L], \mathcal{A}_{ExtSGD}) \quad \leq \quad \gamma_t(\mathcal{F}_{str}^N[\mu], \mathcal{A}_{SGD}), \tag{19}$$

$$\frac{1}{2}\frac{N}{\mu^2 t}(1 - \mathcal{O}((\ln t)/t)) \leq \gamma_t(\mathcal{F}_{sm}^N[\mu, L], \mathcal{A}_{ExtSGD}) \quad \leq \quad \gamma_t(\mathcal{F}_{esm}^N[\mu, L], \mathcal{A}_{SGD})$$

$$\leq \quad \frac{16N}{\mu^2 t}(1 + \mathcal{O}(1/t)). \tag{20}$$

We conclude the following observations (our contributions):

1. The first inequality (18) is from [1]. Comparing (19) to (18) shows that as a lower bound for $\gamma_t(\mathcal{F}_{str}^N[\mu], \mathcal{A}_{SGD})$ (SGD for the class of strongly convex objective functions) our lower bound (17) is dimension independent and improves the lower bound (13) of [1] by a factor $775 \cdot d$. This is a significant improvement.

2. However, our lower bound does not hold for the larger class $\mathcal{A}_{stoch}$. This teaches us that if we wish to reach smaller (better) expected convergence rates, then one approach is to step beyond SGD where our lower bound does not hold implying that within $\mathcal{A}_{stoch}$ there may be an opportunity to find an algorithm leading to at most a factor $32 \cdot 775 \cdot d$ smaller expected convergence rate compared to upper bound (20). This is the first exact quantification into what extent a better (practical) algorithm when compared to classical SGD can be found. E.g., Adam [10], AdaGrad [6], SGD-Momentum [23], RMSProp [24] are all in $\mathcal{A}_{stoch}$ and can beat classical SGD by at most a factor $32 \cdot 775 \cdot d$.

3. When searching for a better algorithm in $\mathcal{A}_{stoch}$ which significantly improves over SGD, it does not help to take an SGD-like algorithm which uses step sizes that are a function of iteratively computed estimates of $\nabla F(w_t)$ and $Y_t$ as this would keep such an algorithm in $\mathcal{A}_{ExtSGD}$ for which our lower bound is tight.

4. Another approach to reach smaller expected convergence rates is to stick with SGD but consider a smaller restricted class of objective functions for which more/other information in the form of extra global parameters is available for adaptively computing $\eta_t$.

5. For strongly convex and smooth, respectively expected smooth, objective functions the algorithm $A \in \mathcal{A}_{SGD}$ with stepsizes $\eta_t = \frac{2}{\mu t + 4L}$, respectively $\eta_t = \frac{2t+1}{(t+1)^2 \mu}$ for $t > \frac{4L}{\mu}$ and $\eta_t = \frac{1}{2L}$ for $t \leq \frac{4L}{\mu}$, realizes the upper bound in (20) for all $t$. Inequalities (20) show that this algorithm is close to optimal: For each $t$, the best sequence of diminishing step sizes which minimizes $Y_t$ can at most achieve a constant (dimension independent) factor $32$ smaller expected convergence rate.

# 4 Lower Bound for Extended SGD

In order to prove a lower bound we propose a specific subclass of strongly convex and smooth objective functions $F$ and we show in the extended SGD setting how, based on recurrence (7), to compute the *optimal* step size $\eta_t$ as a function of $\mu$ and $L$ and an oracle $\mathcal{U}$ with access to $N$, $\nabla F(w_t)$, and $Y_t$, i.e., this step size achieves the smallest $Y_{t+1}$ at the $t$-th iteration.

We consider the following class of objective functions $F$: We consider a multivariate normal distribution of a $d$-dimensional random vector $\xi$, i.e., $\xi \sim \mathcal{N}(m, \Sigma)$, where $m = \mathbb{E}[\xi]$ and $\Sigma = \mathbb{E}[(\xi - m)(\xi - m)^{\mathrm{T}}]$ is the (symmetric positive semi-definite) covariance matrix. The density function of $\xi$ is chosen as

$$g(\xi) = \frac{\exp(\frac{-(\xi - m)^{\mathrm{T}} \Sigma^{-1} (\xi - m)}{2})}{\sqrt{(2\pi)^d |\Sigma|}}.$$

We select component functions $f(w; \xi) = s(\xi) \frac{\|w - \xi\|^2}{2}$, where function $s(\xi)$ is constructed *a-priori* according to the following random process:

- With probability $1 - \mu/L$, we draw $s(\xi)$ from the uniform distribution over interval $[0, \mu/(1 - \mu/L)]$.
- With probability $\mu/L$, we draw $s(\xi)$ from the uniform distribution over interval $[0, L]$.

The following theorem analyses the sequence of optimal step sizes for our class of objective functions and gives a lower bound on the corresponding expected convergence rates. The theorem states that we cannot find a better sequence of step sizes. In other words without any more additional information about the objective function (beyond $\mu, L, N, Y_0, \ldots, Y_t$ for computing $\eta_t$), we can at best prove a general upper bound which is at least the lower bound as stated in the theorem. The proof of the lower bound is presented in the supplementary material:

**Theorem 1.** *We assume that component functions $f(w; \xi)$ are constructed according to the recipe described above with $\mu < L/18$. Then, the corresponding objective function is $\mu$-strongly convex and the component functions are $L$-smooth and convex.*

*If we run Algorithm 1 and assume that access to an oracle $\mathcal{U}$ with access to $N$, $\nabla F(w_t)$, and $Y_t$ is given at the $t$-th iteration (our extended SGD problem setting), then an exact expression for the optimal sequence of stepsizes $\eta_t$ based on $\mu, L, N, Y_0, \ldots, Y_t$ can be given, i.e., this sequence of stepsizes achieves the smallest possible $Y_{t+1}$ at the $t$-th iteration for all $t$. For this sequence of stepsizes,*

$$Y_t \geq \frac{N}{2\mu} \frac{1}{\mu t + 2\mu \ln(t + 1) + W}, \tag{21}$$

*where*

$$W = \frac{L^2}{12(L - \mu)}.$$

In the supplementary material we show numerical experiments in agreement with the presented theorem.

# 5 Conclusion

We have studied the convergence of SGD by introducing a framework for comparing upper bounds and lower bounds and by proving a new lower bound based on straightforward calculus. The new lower bound is dimension independent and improves a factor $775 \cdot d$ over previous work [1] applied to SGD, shows the optimality of step sizes in [17, 7], and shows that practical algorithms like Adam [10], AdaGrad [6], SGD-Momentum [23], RMSProp [24] for strongly convex objective functions can at most achieve a factor $32 \cdot 775 \cdot d$ smaller expected convergence rate compared to classical SGD.

# Acknowledgement

We thank the reviewers for useful suggestions to improve the paper. Phuong Ha Nguyen and Marten van Dijk were supported in part by AFOSR MURI under award number FA9550-14-1-0351.

## Footnotes

[1]We notice that even though stochastic gradient is referred to as SG in literature, the term stochastic gradient descent (SGD) has been widely used in many important works of large-scale learning.

[2]This means that distribution $\mathcal{D}$ in [7] must be over unit vectors $v \in [0, \infty)^n$, where $n$ is the number of component functions, i.e., $n$ possible values for $\xi$. Arbitrary distributions $\mathcal{D}$ correspond to SGD with mini-batches where each component function indexed by $\xi$ is weighted with $v_\xi$.

[3]The bounded gradient assumption, where $\mathcal{S}$ is unbounded, is in conflict with assuming strong convexity as explained in [17].

[4]We also discuss the underlying assumption of convex Lipschitz and show that in order for the analysis in [1] to follow through one – likely tedious but believable – statement still needs a formal proof.

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
