[Supplementary Material · SGD_Tightness_NIPS2019_FULL.pdf]

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

## Supplementary Material

## A   Proof

We extend Theorem 1 with an upper bound used in our numerical experiments.

**Theorem 1** *We assume that component functions $f(w; \xi)$ are constructed according to the recipe described in Section 4 with $\mu < L/18$. Then, the corresponding objective function is $\mu$-strongly convex and the component functions are L-smooth and convex.*

*If we run Algorithm 1 and assume that access to an oracle $\mathcal{U}$ with access to $N$, $\nabla F(w_t)$, and $Y_t$ is given at the t-th iteration (our extended SGD problem setting), then an exact expression for the optimal sequence of stepsizes $\eta_t$ based on $\mu, L, N, Y_0, \ldots, Y_t$ can be given, i.e., this sequence of stepsizes achieves the smallest possible $Y_{t+1}$ at the t-th iteration for all t. For this sequence of stepsizes,*

$$Y_t \geq \frac{N}{2\mu} \frac{1}{\mu t + 2\mu \ln(t+1) + W},$$

where $W = \frac{L^2}{12(L-\mu)}$ and for $t \geq T' = \frac{20L}{\mu}$,

$$Y_t \leq \frac{16N}{\mu} \frac{1}{\mu t - 16L}. \tag{22}$$

*Proof.* We first restrict oracle $\mathcal{U}$ to only supply information about $N$ and $Y_t$ at the t-th iteration. At the end of this proof we show that our arguments generalize to the more powerful oracle $\mathcal{U}$ which also provides the full gradient $\nabla F(w_t)$ at the t-th iteration.

Clearly, $f(w; \xi)$ is $s(\xi)$-smooth where the maximum value of $s(\xi)$ is equal to $L$. That is, all functions $f(w; \xi)$ are L-smooth (and we cannot claim a smaller smoothness parameter). We notice that

$$\mathbb{E}_\xi[s(\xi)] = (1 - \mu/L)\frac{\mu/(1-\mu/L)}{2} + (\mu/L)\frac{L}{2} = \mu$$

and

$$\mathbb{E}_\xi[s(\xi)^2] = (1 - \mu/L)\frac{(\mu/(1-\mu/L))^2}{12} + (\mu/L)\frac{L^2}{12}$$

$$= \frac{\mu(L + \frac{\mu}{1-\mu/L})}{12} = \frac{\mu L^2}{12(L-\mu)}.$$

With respect to $f(w; \xi)$ and distribution $g(\xi)$ we define

$$F(w) = \mathbb{E}_\xi[f(w; \xi)] = \mathbb{E}_\xi[s(\xi)\frac{\|w - \xi\|^2}{2}].$$

*Since $s(\xi)$ only assigns a random variable to $\xi$ which is drawn from a distribution whose description is not a function of $\xi$, random variables $s(\xi)$ and $\xi$ are statistically independent. Therefore, $F(w) =$*

$$\mathbb{E}_\xi[s(\xi)\frac{\|w - \xi\|^2}{2}] = \mathbb{E}_\xi[s(\xi)]\mathbb{E}_\xi[\frac{\|w - \xi\|^2}{2}] = \mu\mathbb{E}_\xi[\frac{\|w - \xi\|^2}{2}]$$

Notice:

1. $\|w - \xi\|^2 = \|(w - m) + (m - \xi)\|^2 = \|w - m\|^2 + 2\langle w - m, m - \xi\rangle + \|m - \xi\|^2$.

2. Since $m = \mathbb{E}[\xi]$, we have $\mathbb{E}[m - \xi] = 0$.

3. $\mathbb{E}[\|m - \xi\|^2] = \sum_{i=1}^d \mathbb{E}[(m_i - \xi_i)^2] = \sum_{i=1}^d \Sigma_{i,i} = \text{Tr}(\Sigma)$.

Therefore, $F(w) = \mu\mathbb{E}_\xi[\frac{\|w-\xi\|^2}{2}] = \mu\frac{\|w-m\|^2}{2} + \mu\frac{\text{Tr}(\Sigma)}{2}$, and this shows $F$ is $\mu$-strongly convex and has minimum $w_* = m$.

Since

$$\nabla_w[\|w - \xi\|^2] = \nabla_w[\langle w, w \rangle - 2\langle w, \xi \rangle + \langle \xi, \xi \rangle]$$
$$= 2w - 2\xi = 2(w - \xi),$$

we have

$$\nabla_w f(w; \xi) = s(\xi)(w - \xi).$$

In our notation

$$N = 2\mathbb{E}_\xi[\|\nabla f(w_*; \xi)\|^2] = 2\mathbb{E}_\xi[s(\xi)^2\|w_* - \xi\|^2].$$

By using similar arguments as used above we can split the expectation and obtain

$$N = 2\mathbb{E}_\xi[s(\xi)^2\|w_* - \xi\|^2] = 2\mathbb{E}_\xi[s(\xi)^2]\mathbb{E}_\xi[\|w_* - \xi\|^2].$$

We already calculated ($w_* = m$)

$$\mathbb{E}_\xi[\|w_* - \xi\|^2] = \|w_* - m\|^2 + \text{Tr}(\Sigma) = \text{Tr}(\Sigma)$$

and we know

$$\mathbb{E}_\xi[s(\xi)^2] = \frac{\mu L^2}{12(L - \mu)}.$$

This yields

$$N = 2\mathbb{E}_\xi[s(\xi)^2]\mathbb{E}_\xi[\|w_* - \xi\|^2] = \frac{\mu L^2}{6(L - \mu)}\text{Tr}(\Sigma).$$

In the SGD algorithm we compute

$$w_{t+1} = w_t - \eta_t \nabla f(w_t; \xi_t)$$
$$= w_t - \eta_t s(\xi_t)(w_t - \xi_t)$$
$$= (1 - \eta_t s(\xi_t))w_t + \eta_t s(\xi_t)\xi_t.$$

We draw $\xi$ from its distribution and set $w_0 = \xi$. Therefore,

$$Y_0 = \mathbb{E}[\|w_0 - w_*\|^2] = \mathbb{E}[\|\xi - w_*\|^2] = \text{Tr}(\Sigma).$$

Let $\mathcal{F}_t = \sigma(w_0, \xi_0, \ldots, \xi_{t-1})$ be the $\sigma$-algebra generated by $w_0, \xi_0, \ldots, \xi_{t-1}$. We derive $\mathbb{E}[\|w_{t+1} - w_*\|^2|\mathcal{F}_t]$

$$= \mathbb{E}[\|(1 - \eta_t s(\xi_t))(w_t - w_*) + \eta_t s(\xi_t)(\xi_t - w_*)\|^2|\mathcal{F}_t]$$

which is equal to

$$\mathbb{E}[(1 - \eta_t s(\xi_t))^2\|w_t - w_*\|^2$$
$$+ 2\eta_t s(\xi_t)(1 - \eta_t s(\xi_t))\langle w_t - w_*, \xi_t - w_* \rangle$$
$$+ \eta_t^2 s(\xi_t)^2\|\xi_t - w_*\|^2|\mathcal{F}_t]. \tag{23}$$

Given $\mathcal{F}_t$, $w_t$ is not a random variable. Furthermore, we can use linearity of taking expectations and as above split expectations:

$$\mathbb{E}[(1 - \eta_t s(\xi_t))^2]\|w_t - w_*\|^2$$
$$+ \mathbb{E}[2\eta_t s(\xi_t)(1 - \eta_t s(\xi_t))]\langle w_t - w_*, \mathbb{E}[\xi_t - w_*]\rangle$$
$$+ \mathbb{E}[\eta_t^2 s(\xi_t)^2]\mathbb{E}[\|\xi_t - w_*\|^2]. \tag{24}$$

Again notice that $\mathbb{E}[\xi_t - w_*] = 0$ and $\mathbb{E}[\|\xi_t - w_*\|^2] = \text{Tr}(\Sigma)$. So, $\mathbb{E}[\|w_{t+1} - w_*\|^2|\mathcal{F}_t]$ is equal to

$$\mathbb{E}[(1 - \eta_t s(\xi_t))^2]\|w_t - w_*\|^2 + \eta_t^2 \frac{N}{2}$$
$$= (1 - 2\eta_t\mu + \eta_t^2 \frac{\mu L^2}{12(L - \mu)})\|w_t - w_*\|^2 + \eta_t^2 \frac{N}{2}$$
$$= (1 - \mu\eta_t(2 - \frac{\eta_t}{12}\frac{L^2}{L - \mu}))\|w_t - w_*\|^2 + \eta_t^2 \frac{N}{2}.$$

In terms of $Y_t = \mathbb{E}[\|w_t - w_*\|^2]$, by taking the full expectation (also over $\mathcal{F}_t$) we get

$$Y_{t+1} = (1 - \mu\eta_t(2 - \frac{\eta_t}{12}\frac{L^2}{L-\mu}))Y_t + \eta_t^2\frac{N}{2}. \tag{25}$$

This is very close to recurrence (4).

Equation (25) expresses $Y_{t+1}$ as a function $Y_{t+1}(\eta_t, Y_t)$ of $\eta_t$ and $Y_t$. Given $Y_0$, we want to minimize $Y_{t+1}$ with respect to the step sizes $\eta_t, \eta_{t-1}, \ldots, \eta_0$. For $i < t$ we derive

$$\frac{\partial Y_{t+1}}{\partial \eta_i} = \frac{\partial Y_{t+1}}{\partial Y_t}\frac{\partial Y_t}{\partial \eta_i} = (1 - \mu\eta_t(2 - \frac{\eta_t}{12}\frac{L^2}{L-\mu}))\frac{\partial Y_t}{\partial \eta_i}$$

and for $i = t$ we derive

$$\frac{\partial Y_{t+1}}{\partial \eta_i} = -2\mu Y_t + 2\mu\frac{\eta_t}{12}\frac{L^2}{L-\mu}Y_t + N\eta_t. \tag{26}$$

We reach a stationary point for $Y_{t+1}$ as a function of step sizes $\eta_t, \eta_{t-1}, \ldots, \eta_0$ if each of the partial derivatives with respect to $\eta_i$ is equal to 0. We notice that if for all $t$

$$1 - \mu\eta_t(2 - \frac{\eta_t}{12}\frac{L^2}{L-\mu}) > 0, \tag{27}$$

then, for $i < t$, $\frac{\partial Y_{t+1}}{\partial \eta_i} = 0$ if and only if $\frac{\partial Y_t}{\partial \eta_i} = 0$. This implies that $Y_{t+1}$ has a stationary point if and only if

$$\forall_{0 \le i \le t} \frac{\partial Y_{i+1}}{\partial \eta_i} = 0.$$

Hence, if a step size sequence satisfies this for all $t$, then it leads to stationary points for all $Y_{t+1}$ as function of $\eta_t, \eta_{t-1}, \ldots, \eta_0$. So, such a sequence of step sizes simultaneously achieves stationary points for all $Y_{t+1}$.

For the argument to hold, we need to prove (27). The left hand side of (27) achieves its minimum value

$$1 - 12\mu\frac{L-\mu}{L^2}$$

for $\eta_t = 12\frac{L-\mu}{L^2}$. For $\mu < \frac{L}{12}$, $12\mu(L-\mu) < 12\mu L < L^2$ implying that this minimum value is larger than zero.

As explained above the optimal step size $\eta_t$ in a sequence of optimal step sizes that minimizes all expected convergence rates $Y_t$ is computed by taking the derivative of $Y_{t+1}$ with respect to $\eta_t$. This derivative is equal to (26) and shows that the minimum is achieved for

$$\eta_t = \frac{2\mu Y_t}{N + \frac{\mu L^2}{6(L-\mu)}Y_t} \tag{28}$$

giving, see (25),

$$Y_{t+1} = Y_t - \frac{2\mu^2 Y_t^2}{N + \frac{\mu L^2}{6(L-\mu)}Y_t}$$

$$= Y_t - \frac{2\mu^2 Y_t^2}{N(1 + Y_t/\text{Tr}(\Sigma))}. \tag{29}$$

We note that $Y_{t+1} \le Y_t$ for any $t \ge 0$. We proceed by proving a lower bound on $Y_t$. Clearly,

$$Y_{t+1} \ge Y_t - \frac{2\mu^2 Y_t^2}{N} \tag{30}$$

Let us define $\gamma = 2\mu^2/N$. We can rewrite (30) as follows:

$$\gamma Y_{t+1} \ge \gamma Y_t(1 - \gamma Y_t) \text{ or}$$
$$(\gamma Y_{t+1})^{-1} \le 1 + (\gamma Y_t)^{-1} + \frac{1}{(\gamma Y_t)^{-1} - 1}. \tag{31}$$

In order to make the inequality above correct, we require $1 - \gamma Y_t > 0$ for any $t \geq 0$. Since $Y_{t+1} \leq Y_t$, we only need $Y_0 < \frac{1}{\gamma}$. This is implied by $Y_0 = \text{Tr}(\Sigma) < \frac{2}{3\gamma}$, a condition which is needed in the next sequence of arguments. This stronger condition means that we need

$$\text{Tr}(\Sigma) < \frac{N}{3\mu^2}, \text{ i.e., } \text{Tr}(\Sigma) < \frac{\mu L^2}{6(L-\mu)} \frac{\text{Tr}(\Sigma)}{3\mu^2}$$

after substituting $N$. This is equivalent to $\mu < \frac{L^2}{18(L-\mu)}$ which is true for $\mu < L/18$.

By using induction on $t$, upper bound (31) implies

$$(\gamma Y_{t+1})^{-1} \leq (t+1) + (\gamma Y_0)^{-1} + \sum_{i=0}^{t} \frac{1}{(\gamma Y_i)^{-1} - 1}. \tag{32}$$

In order to further upper bound the sum in the right hand side, we first find a lower bound on $(\gamma Y_i)^{-1}$. We rewrite equation (29) as

$$(\gamma Y_{t+1}) = (\gamma Y_t)(1 - \frac{(\gamma Y_t)}{1 + Y_t/\text{Tr}(\Sigma)}).$$

Since $Y_t \leq Y_0 = \text{Tr}(\Sigma)$, we have

$$(\gamma Y_{t+1}) \leq (\gamma Y_t)(1 - \frac{(\gamma Y_t)}{2}).$$

This translates into

$$
\begin{aligned}
(\gamma Y_{t+1})^{-1} &\geq \frac{(\gamma Y_t)^{-1}}{1 - (\gamma Y_t)/2} = \frac{(\gamma Y_t)^{-2}}{(\gamma Y_t)^{-1} - 1/2} \\
&= \frac{1}{2} + (\gamma Y_t)^{-1} + \frac{1}{4(\gamma Y_t)^{-1} - 2} \\
&\geq \frac{1}{2} + (\gamma Y_t)^{-1},
\end{aligned}
$$

where the last inequality follows from $(\gamma Y_t)^{-1} > (\gamma Y_0)^{-1} = (\gamma \text{Tr}(\Sigma))^{-1} > 1$ making $4(\gamma Y_t)^{-1} - 2$ positive.

The resulting inequality leads to a recurrence and by using induction on $t$ we obtain

$$(\gamma Y_{t+1})^{-1} \geq (t+1)/2 + (\gamma Y_0)^{-1}.$$

Now we are able to upper bound

$$
\begin{aligned}
\sum_{i=0}^{t} \frac{1}{(\gamma Y_i)^{-1} - 1} &\leq \sum_{i=0}^{t} \frac{1}{i/2 + (\gamma Y_0)^{-1} - 1} \\
&= 2 \sum_{i=0}^{t} \frac{1}{i + 2((\gamma Y_0)^{-1} - 1)}.
\end{aligned}
$$

We showed earlier that $\mu < L/18$ implies $Y_0 < \frac{2}{3\gamma}$. Substituting this upper bound in our derivation leads to

$$\sum_{i=0}^{t} \frac{1}{(\gamma Y_i)^{-1} - 1} \leq 2 \sum_{i=0}^{t} \frac{1}{i + 1} \leq 2\ln(t+2).$$

Combining with (32) we have the following inequality:

$$(\gamma Y_{t+1})^{-1} \leq (t+1) + (\gamma Y_0)^{-1} + 2\ln(t+2).$$

Reordering, substituting $\gamma = 2\mu^2/N$, and replacing $t+1$ by $t$ yields, for $t \geq 0$, the lower bound

$$
\begin{aligned}
Y_t &\geq \frac{N}{2\mu} \frac{1}{\mu t + N/(2\mu Y_0) + 2\mu \ln(t+1)} \\
&= \frac{N}{2\mu} \frac{1}{\mu t + 2\mu \ln(t+1) + W},
\end{aligned}
$$

where

$$W = N/(2\mu Y_0) = \frac{L^2}{12(L-\mu)}.$$

We now extend oracle $\mathcal{U}$ to also provide information about *full gradient* $\nabla F(w_t)$ at the $t$-th iteration. The above proof generalizes to this more powerful oracle. This is because of the reason why we are allowed to transform (23) into (24), i.e., $\eta_t$ and $\xi_t$ must be independent to get (24) from (23). If the construction of $\eta_t$ does not depend on $\xi_t$ (or $\nabla f(w_t; \xi_t)$), then only $Y_t$ is required to construct the optimal stepsize $\eta_t$. It implies that the information of $\nabla F(w_t)$ is not useful and we can borrow the above proof to arrive at the lower bound of this theorem.

The upper bound for $Y_t$ comes from the following fact. If we run Algorithm 1 with step size $\eta_t' = \frac{2}{\mu t + 4L}$ for $t \geq 0$ in [19], then we have from [19] an expected convergence rate

$$Y_t' \leq \frac{16N}{\mu} \frac{1}{\mu(t - T') + 4L}$$

for $t \geq T'$, where

$$T' = \frac{4L}{\mu} \max\{\frac{L\mu Y_0}{N}, 1\} - \frac{4L}{\mu}.$$

Substituting

$$N = \frac{\mu L^2}{6(L-\mu)} \text{Tr}(\Sigma) \text{ and } Y_0 = \text{Tr}(\Sigma)$$

yields $T' \leq \frac{20L}{\mu}$. Since $\eta_t$ is the most optimal step size and $\eta_t'$ is not, $Y_t \leq Y_t'$. I.e., we have for $t \geq \frac{20L}{\mu} \geq T'$,

$$Y_t \leq \frac{16N}{\mu} \frac{1}{\mu(t - \frac{20L}{\mu}) + 4L} = \frac{16N}{\mu} \frac{1}{\mu t - 16L}.$$

$\square$

## B   Numerical Experiments

We verify our theory by considering simulations with different values of sample size $n$ (1000, 10000, and 100000) and vector size $d$ (10, 100, and 1000). We generate $m \in \mathbb{R}^d$ and a diagonal matrix $\Sigma \in \mathbb{R}^{d \times d}$ by drawing each element in $m$ and each element on the diagonal of $\Sigma$ at random from a uniform distribution over $[0, 1]$. We have $L = 1$ and $\mu = 1/n$ where $n$ is the number of samples. Hence the condition number $L/\mu$ is equal to $n$ and represents the number of SGD iterations in a single epoch. We experimented with 10 runs and reported the average results.

We denote the labels "Upper Y_t" (red line) and "Lower Y_t" (violet line) in Figure 1 as the upper and lower bounds of $Y_t$ in (22) and (21) respectively (with a vertical line at epoch 20 because we expect to see the upper bound take effect when $t \geq T' = 20L/\mu$, see supplemental material A); "Y_t_opt" (orange line) as $Y_t$ defined in Theorem 1 computed by using information from oracle $\mathcal{U}$; "Y_t" (green line) as the squared norm of the difference between $w_t$ and $w_*$, where $w_t$ is generated from Algorithm 1 with learning rate (28). Note that $Y_t$ in Figure 1 is computed as average of 10 runs of $\|w_t - w_*\|^2$ (not exactly $\mathbb{E}[\|w_t - w_*\|^2]$).

"Upper Y_t" (red line), "Lower Y_t" (violet line) and "Y_t_opt" (orange line) do not oscillate because they can be correctly computed using formulas (22), (21), and (29), respectively, i.e., they have no variation. The green line "Y_t" for stepsize $\eta_t = \frac{2}{\mu t + 4L}$ in Figure 1 oscillates because our analysis does not consider the variance of $\|w_t - w_*\|^2$. From (4) we infer that a decrease in $\eta_t$ leads to a decrease of the variance of $\|w_t - w_*\|^2$. This fact is reflected in all subfigures in Figure 1. We expect that increasing $d$ and $n$ (the number of dimensions in data and the number of data points) will increase the variance. Hence, it requires larger $t$ to make the variance approach 0 as shown in Figure 1. For sufficiently large $t$, the optimality of $\eta_t = \frac{2}{\mu t + 4L}$ is clearly shown in Figure 1 when $n = 1000$ and $d = 10$, i.e., the green line is in between red line (upper bound) and violet line (lower bound). We note that "Lower Y_t" and "Y_t_opt" are very close to each other in Figure 1 and the difference between them is shown in Figure 2.

Figure 1: $Y_t$ and its upper and lower bounds

Figure 2: The difference between "Lower Y_t" and "Y_t_opt" ($n = 10000$, $d = 100$)

## C  Related Work

In [1], the authors showed that the lower bound of $Y_t$ is $\mathcal{O}(1/t)$ *with* bounded gradient assumption for objective function $F$ over a convex set $\mathcal{S}$. To show the lower bound, the authors use the following three assumptions for the objective function $F$:

1. The assumption of a strongly convex objective function, i.e., Assumption 1 (see Definition 3 in [1]).

2. There exists a bounded convex set $\mathcal{S} \subset \mathbb{R}^d$ such that

$$\mathbb{E}[\|\nabla f(w;\xi)\|^2] \leq \sigma^2$$

   for all $w \in \mathcal{S} \subset \mathbb{R}^d$ (see Definition 1 in [1]). Notice that this is not the same as the bounded gradient assumption where $\mathcal{S} = \mathbb{R}^d$ is unbounded.

3. The objective function $F$ is a convex Lipschitz function, i.e., there exists a positive number $K$ such that
$$\|F(w) - F(w')\| \leq K\|w - w'\|, \forall w, w' \in \mathcal{S} \subset \mathbb{R}^d.$$
   We notice that this assumption actually implies the assumption on bounded gradients as stated above.

**On the existence of the assumption of bounded convex set $\mathcal{S} \subset \mathbb{R}^d$ where SGD converges:** let us restate the example in [19], i.e. $F(w) = \frac{1}{2}(f_1(w) + f_2(w))$ where $f_1(w) = \frac{1}{2}w^2$ and $f_2(w) = w$. It is obvious that $F$ is strongly convex but $f_1$ and $f_2$ are not. Let $w_0 = 0 \in \mathcal{S}$, for any number $t \geq 0$, with probability $\frac{1}{2^t}$, the steps of SGD algorithm for all $i < t$ are $w_{i+1} = w_i - \eta_i$. This implies that $w_t = -\sum_{i=1} \eta_i$. Since $\sum_{i=1} \eta_i = \infty$, $w_t$ will escape the set $\mathcal{S}$ when $t$ is sufficiently large. We conclude that in $\mathcal{F}_{str}$ there are objective functions that can escape any bounded set $\mathcal{S}$ with non-zero probability.

If $\mathcal{S}$ is $\mathbb{R}^d$, we have the following results:

**On the non-coexistence of the assumption of a bounded gradient over $\mathbb{R}^d$ and assumption of having strong convexity:** As pointed out in [19], the assumption of bounded gradient does not co-exist with strongly convex assumption. As shown in [17, 3], Assumption 1 on strong convexity implies

$$2\mu[F(w) - F(w_*)] \leq \|\nabla F(w)\|^2 , \, \forall w \in \mathbb{R}^d. \tag{33}$$

As shown in [19], for any $w \in \mathbb{R}^d$, we have

$$2\mu[F(w) - F(w_*)] \overset{(33)}{\leq} \|\nabla F(w)\|^2 = \|\mathbb{E}[\nabla f(w; \xi)]\|^2$$
$$\leq \mathbb{E}[\|\nabla f(w; \xi)\|^2] \leq \sigma^2.$$

Therefore,

$$F(w) \leq \frac{\sigma^2}{2\mu} + F(w_*), \forall w \in \mathbb{R}^d.$$

Note that, the from Assumption 1 and $\nabla F(w_*) = 0$, we have

$$F(w) \geq \mu \|w - w_*\|^2 + F(w_*), \forall w \in \mathbb{R}^d.$$

Clearly, the two last inequalities contradict to each other for sufficiently large $\|w - w_*\|^2$. Precisely, only when $\sigma$ is equal to $\infty$, then the assumption of bounded gradient and the assumption of strongly convexity of $F$ can co-exist. However, $\sigma$ cannot be $\infty$ and this result implies that there does not exist any objective function $F$ satisfies the assumption of bounded gradients over $\mathbb{R}^d$ and the assumption of having a strongly convex objective function at the same time.

**On the non-coexistence of the assumption of being convex Lipschitz over $\mathbb{R}^d$ and assumption of being strongly convex:** Moreover, we can also show that the assumption of convex Lipschitz function does not co-exist with the assumption of being strongly convex. As shown in Section 2.3 in [1], the assumption of Lipschitz function implies that $\|\nabla F(w)\| \leq K, \forall w \in R^d$. Hence, by using the same argument from the analysis of the non-coexistence of bounded gradient assumption and assumption of strongly convex, we can conclude that these two assumptions cannot co-exist. In other words, there does not exist an objective function $F$ which satisfies the assumption of convex Lipschitz function and assumption of being strongly convex at the same time.

### C.1 Discussion on the usage of Assumptions in [1]

As stated in Section 3 and Section 4.1.1 in [1], the authors construct a class of strongly convex Lipschitz objective function $F$ which has $K = \sigma$. The authors showed that the problem of convex optimization for the constructed class of objective functions $F$ is at least as hard as estimating the biases of $d$ independent coins (i.e., the problem of estimating parameters of Bernoulli variables). As one additional important assumption to prove the lower bound of a first order stochastic algorithm, the authors assume the **existence** of stepsizes $\eta_t$ which make an first order stochastic algorithm converge for a given objective function $F$ under the three aforementioned assumptions (see Lemma 2 in [1]). Note that the proof of the lower bound of $Y_t$ is described in Theorem 2 in [1] and Theorem 2 uses their Lemma 2. If their Lemma 2 is not valid, then the proof of the lower bound of $Y_t$ in Theorem 2 is also not valid.

Given the proof strategy in [1] of the convergence of a first order stochastic algorithm, one may require that the convex set $\mathcal{S}$ where $F$ has all these nice properties must be $\mathbb{R}^d$ as explained above. This, however, will lead to the non-coexistence of bounded gradient assumption and strongly convex assumption and the non-coexistence of Lipschitz function assumption and strongly convex assumption as discussed above. In this case, their Lemma 2 is not valid because of non-existence of an objective function $F$, in which case the proof of lower bound of $Y_t$ in Theorem 2 is not correct.

However, we explain why the setup as proposed in [1] may still be acceptable and lead to a proper lower bound: The paper assumes that we only restrict the analysis of SGD in a bounded convex set $\mathcal{S}$ which is not $\mathbb{R}^d$, and only in this bounded set $\mathcal{S}$ we assume that objective function acts like a Lipschitz function (implying bounded gradients in $\mathcal{S}$).

There are two possible cases at the $t$-th iteration a first order stochastic algorithm, the algorithm diverges or converges. Let us define $p_t = Pr(w_t \notin \mathcal{S})$. Hence, $Pr(w_t \in \mathcal{S}) = 1 - p_t$. Let

$$Y_t^{conv} = \mathbb{E}[\|w_t - w_*\|^2 | w_t \in \mathcal{S}]$$

and

$$Y_t^{div} = \mathbb{E}[\|w_t - w_*\|^2 | w_t \notin \mathcal{S}].$$

Since $Y_t = \mathbb{E}[\|w_t - w_*\|^2$, $Y_t$ is equal to

$$
\begin{aligned}
Y_t &= p \cdot Y_t^{div} + (1 - p) \cdot Y_t^{conv} \\
&\geq p \cdot Y_t^{conv} + (1 - p) \cdot Y_t^{conv} \\
&\geq Y_t^{conv} \\
&\geq \text{ lower bound in [1].}
\end{aligned}
$$

The above derivation hinges on the first inequality where we assume $Y_t^{div} \geq Y_t^{conv}$. Typically, for strongly convex objective functions and $w_*, w_0 \in \mathcal{S}$), it seems always true that $Y_t^{div} \geq Y_t^{conv}$ because $w_t$ gets far from $w_*$ for the divergence case and it gets close to $w_*$ for the convergence case. Of course a proper proof of this property is still needed in order to rigorously complete the argument leading to the lower bound in [1]. In fact this remains an open problem (one can invent strange corner cases that need extra thought/proof).

The above result is interesting because now we **only** need to prove the convergence of a first order stochastic algorithm in a certain convex set $\mathcal{S}$ with a certain probability $p$. This is completely different from the proof of convergence of e.g. SGD in the general case as in [15] and [19, 8] where we need to prove it with probability 1.

## C.2 Setup

We describe the setup of the class of strong convex functions proposed in [1].

As shown in Section 4.1.1 [1], the following two sets are required.

1. Subset $\mathcal{V} \subset \{-1, +1\}^d$ and $\mathcal{V} = \{\alpha^1, \dots, \alpha^M\}$ with $\Delta_H(\alpha^j, \alpha^k) \geq \frac{d}{4}$ for all $j \neq k$, where $\Delta_H$ denotes the Hamming metric, i.e $\Delta_H(\alpha, \beta) := \sum_{i=1}^d \mathbb{I}[\alpha_i \neq \beta_i]$. As discussed by the authors, $|\mathcal{V}| = M \geq (2/\sqrt{e})^{\frac{d}{2}}$.
2. Subset $\mathcal{F}_{base} = \{f_i^+, f_i^-, i = 1, \dots, d\}$ where $f_i^+, f_i^-$ will be designed depending on the problem at hand.

Given $\mathcal{V}, \mathcal{F}_{base}$ and a constant $\delta \in (0, \frac{1}{4}]$, we define the function class $\mathcal{F}(\delta) := \{F_\alpha, \alpha \in \mathcal{V}\}$ where

$$F_\alpha(w) := \frac{c}{d} \sum_{i=1}^d \{(1/2 + \alpha_i \delta) f_i^+(w) + (1/2 - \alpha_i \delta) f_i^-(w)\}. \tag{34}$$

The $\mathcal{F}_{base}$ and constant $c$ are chosen in such a way that $\mathcal{F}(\delta) \subset \mathcal{F}$ where $\mathcal{F}$ is the class of strongly convex objective functions defined over set $\mathcal{S}$ and satisfies all the assumptions as mentioned before. In case $\mathcal{F}$ is the class of strongly convex functions, the key idea to compute the lower bound of SGD proposed in [1] by applying Fano's inequality [26] and Le Cam's bound [5, 14] is as follows: If an SGD algorithm $\mathcal{M}_t$ works well for optimizing a given function $F_{\alpha^*}, \alpha^* \in \mathcal{V}$ with a given oracle $\mathcal{U}$, then there exists a hypothesis test finding $\hat{\alpha}$ such that:

$$\frac{1}{3} \geq \text{Pr}_\mathcal{U}[\hat{\alpha}(\mathcal{M}_t) \neq \alpha] \geq 1 - 2\frac{16dt\delta^2 + \log(2)}{d \log(2/\sqrt{e})}. \tag{35}$$

From (35), we have

$$\frac{16dt\delta^2 + \log(2)}{d \log(2/\sqrt{e})} \approx \frac{16dt\delta^2}{d \log(2/\sqrt{e})} \geq 2/3.$$

Hence,

$$t \geq \frac{\log(2/\sqrt{e})}{48} \frac{1}{\delta^2}. \tag{36}$$

As shown in Section 4.3 [1], to proceed the proof, we set $Y_t = \frac{c\delta^2 r^2}{18(1-\theta)}$. Combining with (36) yields

$$Y_t \geq \frac{1}{t} \frac{\log(2/\sqrt{e})}{864} \frac{cr^2}{1-\theta}. \tag{37}$$

In addition to the proof of the lower bound, we also need to set $c = \frac{Ld}{rd^{1/p}}$ and $\mu^2 = \frac{L}{rd^{1/p}}(1-\theta)$ where $\mathcal{S} = \mathrm{B}_\infty(r)$. By substituting $c$ and $\mu^2$ into (37), we obtain:

$$Y_t \geq \frac{1}{t} \frac{\log(2/\sqrt{e})}{864d} \frac{1}{\mu^2} c^2 r^2. \tag{38}$$

To complete the description of the setup in [1], we briefly describe the proposed oracle $\mathcal{U}$ which outputs some information to the SGD algorithm at each iteration for constructing the stepsize $\eta_t$. There are two types of oracle $\mathcal{U}$ defined as follows.

1. Oracle $\mathcal{U}_A$: 1-dimensional unbiased gradients
   (a) Pick an index $i \in 1, \ldots, d$ uniformly at random.
   (b) Draw $b_i \in \{0, 1\}$ according to a Bernoulli distribution with parameter $1/2 + \alpha_i \delta$.
   (c) For the given input $x \in \mathcal{S}$, return the value $f_i$ and subgradient $\nabla f_i$ of the function

   $$f_{i,A} := c[b_i f_i^+ + (1 - b_i)f_i^-].$$

2. Oracle $\mathcal{U}_B$: $d$-dimensional unbiased gradients.
   - For $i = 1, \ldots, d$, draw $b_i \in \{0, 1\}$ according to a Bernoulli distribution with parameter $1/2 + \alpha_i \delta$.
   - For the given input $x \in \mathcal{S}$, return the value $f_i$ and subgradient $\nabla f_i$ of the function

   $$f_{i,B} := \frac{c}{d} \sum_{i=1}^{d} [b_i f_i^+ + (1 - b_i)f_i^-].$$

### C.3 Analysis and Comparison

In this section, we want to compare our lower bound ($\approx \frac{N}{2\mu^2 t}$) with the one in (38) when $t$ is sufficiently large. In order to do this, we need to compute $N = 2\mathbb{E}[\|\nabla f(w^*; \xi)\|^2]$ for the strongly convex function class proposed in [1]. For the strongly convex case, the authors defined the base functions as follows. Given a parameter $\theta \in [0, 1)$, we have

$$f_i^+(w) = r\theta|w_i + r| + \frac{1-\theta}{4}(w_i + r)^2,$$

$$f_i^-(w) = r\theta|w_i - r| + \frac{1-\theta}{4}(w_i - r)^2,$$

where $w = (w_1, \ldots, w_d)$. Let $e_i$ be $1/2 + \alpha_i \delta$. Substituting $e_i$ in (34) yields $F_\alpha(w) = \frac{1}{d}[\sum_{i=1}^{d} f_{\alpha,i}(w)]$ where $f_{\alpha,i}(w) = c[e_i f_i^+(w) + (1 - e_i)f_i^-(w)]$. Due to the construction of $F_\alpha$, the definition of $f_{\alpha,i}(w)$ and the construction of oracle $\mathcal{U}_A$ or oracle $\mathcal{U}_B$, $w^*$ of $F_\alpha$ can be found by finding each $w_i^*$ for each $f_{\alpha,i}(w)$ first. Precisely, we have the following cases:

1. $w_i < -r$: we have
   - $f_{\alpha,i}(w) = -r\theta(w_i+r)e_i + \frac{1-\theta}{4}(w_i+r)^2 e_i - r\theta(w_i-r)(1-e_i) + \frac{1-\theta}{4}(w_i-r)^2(1-e_i)$.
   - $\nabla f_{\alpha,i}(w) = (1-\theta)e_i r - \frac{1+\theta}{2}r + \frac{1-\theta}{2}w_i$.
   - $\nabla f_{\alpha,i}(w) = 0$ at $w_i^{-r} = r[1 - 2e_i + \frac{2\theta}{1-\theta}]$.
2. $-r \leq w_i \leq r$: we have
   - $f_{\alpha,i}(w) = r\theta(w_i+r)e_i + \frac{1-\theta}{4}(w_i+r)^2 e_i - r\theta(w_i-r)(1-e_i) + \frac{1-\theta}{4}(w_i-r)^2(1-e_i)$.
   - $\nabla f_{\alpha,i}(w) = (1+\theta)e_i r - \frac{1+\theta}{2}r + \frac{1-\theta}{2}w_i$.
   - $\nabla f_{\alpha,i}(w) = 0$ at $w_i^{[-r,r]} = r\frac{1+\theta}{1-\theta}(1 - 2e_i)$.

3. $r \le w_i \le \infty$: we have

- $f_{\alpha,i}(w) = r\theta(w_i+r)e_i + \frac{1-\theta}{4}(w_i+r)^2 e_i + r\theta(w_i-r)(1-e_i) + \frac{1-\theta}{4}(w_i-r)^2(1-e_i)$.
- $\nabla f_{\alpha,i}(w) = (1-\theta)e_i r + \frac{3\theta-1}{2}r + \frac{1-\theta}{2}w_i$.
- $\nabla f_{\alpha,i}(w) = 0$ at $w_i^r = r[1 - 2e_i - 2\frac{\theta}{1-\theta}]$.

Now, we have five important points $w_i^{-r}, w_i^{[-r,r]}, w_i^r, -r$ and $r$ and at these points $F_\alpha$ can be minimum. We consider the following cases

1. $\alpha_i = -1$ and then $e_i = \frac{1}{2} + \alpha_i\delta = \frac{1}{2} - \delta$ where $\delta \in [0, 1/4)$, we have

   - $w_i^{-r} = r[\frac{2\theta}{1-\theta} + 2\delta] > -r$.
   - $w_i^{[-r,r]} = r\frac{1+\theta}{1-\theta}(2\delta)$. In this case $w_i^{[-r,r]}$ may belong $[-r, r]$ or it may be greater than $r$.
   - $w_i^r = r(2\delta - \frac{2\theta}{1-\theta}) < r$ .

   This result implies $F_\alpha$ is minimum at $w_i^* = r$ and $\nabla f_{\alpha,i}(w^*) = cr[(1-\theta)e_i + \theta] = cr[(1-\theta)(1/2 - \delta) + \theta]$. Or it can be minimum at $w_i^{[-r,r]}$ if $w_i^{[-r,r]} \in [-r, r]$ and $\nabla f_{\alpha,i}(w^*) = 0$.

2. $\alpha_i = +1$ and then $e_i = \frac{1}{2} + \alpha_i\delta = \frac{1}{2} + \delta$ where $\delta \in [0, 1/4)$, we have

   - $w_i^{-r} = r[\frac{2\theta}{1-\theta} - 2\delta]$. Since $\frac{2\theta}{1-\theta} - 2\delta > -1$ when $\delta \in [0, 1/4)$ and $\theta \in [0, 1)$. Hence $w_i^{-r} > -r$.
   - $w_i^{[-r,r]} = r\frac{1+\theta}{1-\theta}(-2\delta) < 0$. In this case $w_i^{[-r,r]}$ may belong $[-r, r]$ or it may be smaller than $-r$.
   - $w_i^r = r(-2\delta - \frac{2\theta}{1-\theta}) < r$.

   This result implies $F_\alpha$ is minimum at $w_i^* = -r$ and $\nabla f_{\alpha,i}(w^*) = cr[(1-\theta)e_i - 1] = cr[(1-\theta)(1/2 + \delta) - 1]$. Or it can be minimum at $w_i^{[-r,r]}$ if $w_i^{[-r,r]} \in [-r, r]$ and $\nabla f_{\alpha,i}(w^*) = 0$.

By definition, we have

$$N = 2\mathbb{E}[\|\nabla f_i(w^*)\|^2] = 2\frac{1}{d}\sum_{i=1}^{d}[e_i\|c\nabla f_i^+(w^*)\|^2 + (1-e_i)\|c\nabla f_i^-(w^*)\|^2]$$

From the analysis above, we have four possible $w_i^*$, i.e., $-r, r, r\frac{1+\theta}{1-\theta}(-2\delta)$ and $r\frac{1+\theta}{1-\theta}(2\delta)$. If we plug $w^*$ which has $w_i^* = -r$ or $w_i^* = r$, then we have $[e_i\|c\nabla f_i^+(w^*)\|^2 + (1-e_i)\|c\nabla f_i^-(w^*)\|^2] = (1/2-\delta)c^2r^2$. For $w_i^*$ which has $w_i^* = r\frac{1+\theta}{1-\theta}(-2\delta)$ or $r\frac{1+\theta}{1-\theta}(2\delta)$, we have $[e_i\|c\nabla f_i^+(w^*)\|^2 + (1-e_i)\|c\nabla f_i^-(w^*)\|^2] = (1/4 - \delta^2)(1+\theta)^2c^2r^2$. This proves that

$$N = 2\beta c^2 r^2$$

with $\beta$ somewhere in the range

$$[(\frac{1}{2} - \delta), (\frac{1}{4} - \delta^2)(1+\theta)^2] \text{ or } [(\frac{1}{4} - \delta^2)(1+\theta)^2, (\frac{1}{2} - \delta)],$$

where $\delta \in [0, 1/4)$ and $\theta \in [0, 1)$.

Substituting $N = 2\beta c^2 r^2$ into (38) yields

$$Y_t \ge \frac{\log(2/\sqrt{e})}{(864 \cdot d)(2\beta)}\frac{N}{\mu^2 t}, \tag{39}$$

which is further minimized by taking

$$\beta = \max\{(\frac{1}{2} - \delta), (\frac{1}{4} - \delta^2)(1+\theta)^2\}.$$

Notice that, given our freedom in choosing $\delta$ and $\theta$, we can minimize $\beta$ as a function of $\delta$ and $\theta$ in order to maximize the lower bound in (39). This gives (in the limit) $\delta = 1/4$ with $\theta \leq 2/\sqrt{3} - 1 = 0.155$ leading to $\beta = 1/4$. This leads to the final lower bound

$$Y_t \geq \frac{\log(2/\sqrt{e})}{432 \cdot d} \frac{N}{\mu^2 t}.$$

Clearly, the lower bound in is much smaller than our lower bound of $\approx \frac{N}{2\mu^2 t}$ when $t$ is sufficiently large. Moreover, this lower bound depends on $1/d$ and it becomes smaller when $d$ increases.