[Reviews · NeurIPS 2019]

Reviewer 1



Pros. + paper is clearly written and easy to follow + contributions are clearly stated and sufficient literature review is provided + result seem to be novel and significant as they give important insight about the impossibility of improving SGD types method for the strongly convex case + the analysis appears to be novel as well, which might have some usage in the construction of other lower bound under different assumptions and frameworks. Cons. - the analysis seems to be too narrow and cover a small subset of algorithms --- I have read the authors feedback.

Reviewer 2



Originality: The lower bound seems to be new for the considered class of problems and stochastic algorithms. It is not clear what is the difference in proving the lower bounds from previous similar works. Quality: The authors should mention more related work on stochastic strongly convex optimization with optimal convergence rates. Clarity: There are a couple of papers on stochastic strongly convex optimization that provide 1/t convergence rate for last iterate. It is not clear why the authors paticularly refer to [8,18]. It is also not clear why the lower bounds on Y_t is more interesting than the objective gap. Significance: It is not clear how the results in this paper can advance our understanding on stochastic strongly convex optimization given so many studies on the upper bounds and lower bounds. I have read the rebuttal.

Reviewer 3



Update: Thank you for the feedback, I have read it as well as other reviews. Compared with the vast literature on obtaining upper bounds on convergence rates of stochastic convex optimization problems, less work has been done towards deriving corresponding lower bounds that depict the fundamental hardness of these problems. This paper aims to fill this gap and proposes a general framework for comparing upper and lower bounds. The framework also suggests potential future research directions for obtaining better convergence rates. In addition, this paper proved, for all round t, a lower bound on the expected convergence rate of SGD over any diminishing step-size sequences when applied to strongly convex problems. This bound shows that the step-size schemes proposed in recent work (Gower et al. 2019, and Nguyen et al. 2018) are optimal within a dimension independent constant factor. By restricting its scope to SGD only, it improves upon the general dimension dependent lower bound in Agarwal et al. (2010) which is for all stochastic first-order algorithms including SGD. The framework proposed in this paper gives a clear comparison of existing results and its own result by incorporating both the different problem settings being studied in each work, and the various families each optimization algorithm belongs to. It formalizes the idea that the hardest objective function to optimize in a larger class of functions is no easier than the hardest objective function in a smaller class and the best algorithm from a larger class of algorithms is no worse than the best algorithm from a smaller class. Thus, to get a better bound, we need to either consider a more restricted problem setting which gives us more information, or use a larger set of algorithms to choose the best one. This framework shall be very useful for later research that seeks the possibility to further improve upper bounds or proves lower bounds for specific problems. The proved lower bound is novel, which not only improves upon the existing work, but also confirms the recent proposed step-size schemes are optimal. The proof of the theorem is very clever. The construction of the function class to be studied when only strong-convexity and smoothness parameters are given is creative, and the way of deriving the optimal step-size sequence is cute. However, I have a minor concern: in Line 344, you first drew \xi, and then chose w_0 as the minimizer of f(w;\xi) to make Y_0 equal to Tr(\Sigma). To me, this either means we are studying a smaller algorithm class in which the initialization parameter is restricted, or we need a few steps to initialize which should be included in the total number of steps t. Thus, in the former case, the algorithm class doesn't include classic SGD in which we can choose any w_0; while in the latter case, the lower bound should be modified to include those initial steps. Could you explain why this won't matter? Typos: 1. into what extend --> into what extent 2. Line 39: "smaller" --> "larger" 3. strongly objective functions --> strongly convex objective functions 4. Line 105: in [8] are equivalent... 5. Line 376: proof generalizes

[Author Response · NeurIPS 2019]

Dear Area Chair and Reviewers,

We appreciate all the reviewers for their careful reviews and valuable comments. We have tried our best to incorporate all
reviewers' suggestions below. We hope our answers address the reviewers' concerns. We recall our major contributions
as follows:

1. Prove the strictly tight and dimension-independent properties of our lower bound. This is the **first** work
proving the **dimension-independence** of the lower bound of SGD.
2. Explain how much faster Adam, AdaGrad, SGD-Momentum, RMSProp can be compared to SGD.
3. Develop a new framework to prove the lower bound of SGD which might be extended to other algorithms.
4. Prove the close-to-optimalilty of step-size schemes in [8,18].

**Reviewer 1.** We thank you for your acceptance of the paper and appreciate all your valuable comments. Since the SGD
algorithm is one of the most basic and efficient first order algorithms, our paper only focuses on SGD.

We will take your advice on extending our work to ADAM, AdaGrad etc. as future work: this will need additional
theory beyond what is presented in this submission in order to discuss and prove the lower bounds for these different
algorithms.

**Reviewer 2.** We appreciate your useful comments on the importance of our work. By highlighting our theoretical
contributions we hope to address all your concerns:

Compared to existing bounds, our lower bound is different in the following points. **Firstly**, it is much tighter and
is dimension-independent. This is the **first** work proving the **dimension-independence** of the lower bound of SGD.
**Secondly**, it explains how much faster Adam, AdaGrad, SGD-Momentum, RMSProp algorithms can be compared to
SGD. This result is not achieved in any previously published works according to our best knowledge. **Thirdly**, it is very
challenging to rigorously prove a tight lower bound. Our proof technique is completely different from the previously
known one in [1] as noted by Reviewer 3. In fact, we (are the first to) explain in Section C.1 in Supplementary Material
why the result/proof in [1] (which is considered as one of the most important works in this research line) is not yet fully
complete (but seems very close to being complete). As mentioned by Reviewer 3, there are very few papers studying
the lower bound on SGD because of its complexity. For studying a lower bound, one approach is to compute lower
bounds of the convergence rates of *all* possible step-size schemes (e.g [1] uses many complex techniques to solve this
problem) while our approach is to find the lower bound of the convergence rate of the optimal step-size scheme among
*all* possible ones (we use a very simple technique coined 'extended SGD'). Since our approach only considers one
(optimal in the sense of 'extended SGD') sequence of step sizes for studying bounds on the convergence rate, it turns
out that our analysis becomes more simple and easier to understand because we only need to resort to basic calculus
arguments. **Finally**, our lower bound allows us to prove the close-to-optimality property of step-size schemes proposed
in [8,18]. Due to all the important contributions in theory, we believe that our lower bound significantly advances our
understanding on the lower bound and upper bound of stochastic strongly convex optimization as you questioned.

We may have missed some important works on upper bounds and will update citations in the revised version of the
paper. We focus on [8,18] because they are the only works on upper bounds which have the same setup as the one in
our work. This allows us to rigorously discuss the close-to-optimality of step-size schemes in [8,18].

We will take your advice and research as future work how to modify our proof technique to develop a lower bound on
objective functions without smoothness: it will need additional theory beyond what is presented in this submission.

**Reviewer 3.** We thank you for your acceptance of the paper and appreciate all your helpful feedback. We hope our
following answers properly address all your concerns.

We apologize for any your inconvenience created by our unclear text. Indeed, we simply choose $w_0 = \xi$ with $\xi$ taken
from its distribution. Then the expectation over the starting point $w_0$ is equal to $Y_0 = E[\|w_0 - w_*\|^2] = E_\xi[\|\xi - w_*\|^2]$
= etc. So, there is no extra computation. We do select $w_0$ according to the distribution of $\xi$ and this means that $w_0$ is not
completely arbitrary (in practice we often start $w_0$ in some region where we feel that our a-priori information indicates
a good start). We will update our current draft to avoid any confusion. We thank you again for your helpful comment.

The main result in Theorem 1 is about the lower bound of SGD. It is proved by developing a class of examples. In
this context it is less important to weaken the condition on $\mu$ (which makes the class of examples larger), i.e., try to
get $\mu < L$ instead of $\mu < L/18$. However, we will consider developing lower bounds for general convex (or even
non-convex settings) for SGD, AdaGrad and Adam using our proof framework (i.e., extended SGD) with $\mu < L$ as
future work as you suggested.

[Meta-Review · NeurIPS 2019]

There was bit of discussion about whether this work will have a significant impact or not, but overall all reviewers are in favor of accepting the work.